# Solid-State NMR for Studying the Structure and Dynamics of Viral Assemblies

**DOI:** 10.3390/v12101069

**Published:** 2020-09-24

**Authors:** Lauriane Lecoq, Marie-Laure Fogeron, Beat H. Meier, Michael Nassal, Anja Böckmann

**Affiliations:** 1Molecular Microbiology and Structural Biochemistry, University of Lyon, 7 Passage du Vercors, CEDEX 07, 69367 Lyon, France; lauriane.lecoq@ibcp.fr (L.L.); marie-laure.fogeron@ibcp.fr (M.-L.F.); 2Physical Chemistry, ETH Zurich, 8093 Zurich, Switzerland; beme@ethz.ch; 3Department of Medicine II/Molecular Biology, Medical Center, University Hospital Freiburg, Albert-Ludwigs-University, 79106 Freiburg, Germany

**Keywords:** structure, solid-state NMR, viral proteins, capsids, membrane proteins

## Abstract

Structural virology reveals the architecture underlying infection. While notably electron microscopy images have provided an atomic view on viruses which profoundly changed our understanding of these assemblies incapable of independent life, spectroscopic techniques like NMR enter the field with their strengths in detailed conformational analysis and investigation of dynamic behavior. Typically, the large assemblies represented by viral particles fall in the regime of biological high-resolution solid-state NMR, able to follow with high sensitivity the path of the viral proteins through their interactions and maturation steps during the viral life cycle. We here trace the way from first solid-state NMR investigations to the state-of-the-art approaches currently developing, including applications focused on HIV, HBV, HCV and influenza, and an outlook to the possibilities opening in the coming years.

## 1. Introduction

Solid-state NMR is a structural biology approach that can provide unique information about the molecular assemblies used by viruses to enter host cells, to establish infection and to ensure that progeny virions are released into the environment. Solid-state NMR can contribute to an understanding of how viral proteins and their assemblies function from a structural point of view, revealing weak spots which can be targeted to fight infection through antiviral therapies or vaccines. Solid-state NMR is in this respect complementary to solution NMR, X-ray crystallography and electron cryo-microscopy (cryo-EM), the other atomic-resolution structural biology methods. All can provide images of proteins at resolutions on the order of 1–10 Å. Considering large assemblies, X-ray crystallography was the first approach used to resolve viral capsid and envelope structures at high resolution, and was pioneered by Michael Rossmann and Stephen C. Harrison on plant viruses [1,2], owing to the enormous numbers of progeny viruses per cell that plants produce. Advances in recombinant DNA as well as cell culture technology opened the way to mammalian viruses [3,4]. This approach has subsequently revealed a wide variety of capsid structures, amongst them capsids from rhinovirus [5], poliovirus [6] and hepatitis B virus (HBV) [7], as other early examples of human pathogens. The typical resolution of the structures is around 2–3 Å, but a higher resolution below 1.5 Å has been reached for several viruses (see www.pdb.org). For X-ray structures, at 2.5 Å, the backbone is well-defined and many, though not all, sidechains are visible; at 1.5 Å, most sidechains are well-defined (and in the correct rotameric state) and some hydrogens including water are resolved [8,9]. While the resolution characterizing X-ray, cryo-EM and NMR structures cannot be directly compared, the numbers that they deliver still have a roughly similar meaning.

Resolving capsid structures by X-ray crystallography is, however, frequently impeded by the difficulties in obtaining highly diffracting crystals from these large, often flexible complexes, and by the requirement for milligram amounts of material for crystallization trials. Much of the available structural information has thus been gathered on the soluble subunits of the capsids, called the capsid or core proteins, which might in the crystal assemble into sub-particulate superstructures such as hexamers. This approximates how capsid proteins build hexagonal lattices, which are the foundation of icosahedral capsids [10,11]. Rossmann, besides using crystallography, was one of the first to recognize that a hybrid approach combining the moderate resolution structures from cryo-EM with high-resolution X-ray structures could allow to reconstitute the entire capsid assembly [12]. EM electron density maps, even at lower resolution, are also helpful since they can be used as an initial phasing model for the X-ray structure determination. Today, this follows standard protocols [13], and in addition to the first crystal structures, this highly successful combination approach has largely shaped our view on a large variety of viral capsids. EM thus had, from the beginning, an important share in structural virology, but was for a long time severely limited by the resolution attainable with the available technology. Still, even low resolution maps yielded in some cases structures; the HBV capsid is one of the first examples of a protein whose fold was determined from cryo-EM, although assisted by numerous biochemically derived constraints [14,15].

With the advent of direct detectors, cryo-EM underwent the “resolution revolution” [16], enabling similarly high resolution as X-ray and NMR which, again for the HBV capsid as example, reached 2.8 Å [17]. This step forward was central for structural virology, since it allowed for the first time to look at authentic viruses with atomic resolution, and to investigate structures of virus-like particles in different maturation states, as documented by numerous structures deposited in the PDB. The impressive pictures thus obtained for many viruses, including human pathogens such as Dengue virus (DENV), ZIKA virus, HBV and Severe Acute Respiratory Syndrome Coronavirus-2 (SARS-CoV-2), many of them from Rossmann and coworkers, have shaped our view of infection. Moreover, the high resolution now allows to resolve the arrangement of proteins in challengingly large unit cells, as shown by the impressive 3.5 Å structure of a giant virus capsid, recently reported by Rossmann′s lab, which comprises numerous different proteins which intimately interact to form the capsid shell [18].

However, obtaining homogeneous samples from virus-replicating cells can be very demanding, owing to the often complex mixture of viral particles in different maturation states that coexist in such cells. Single particles can be picked on electron micrographs, allowing to exclude poorly assembled particles and/or to separate clearly different ones, such as the 180 subunit form (triangulation number *T* = 3) vs. the 240 subunit form (*T* = 4) of the HBV capsid. This approach has also been applied to capsids of duck HBV (DHBV), which in their larger core protein feature an extra domain that can adopt disordered and folded states [19]. Still, the differences must be large enough to enable separation of the different particle states, otherwise the resulting heterogeneity will manifest itself as a loss of resolution, and tracking different states remains difficult if not impossible.

This is where solid-state NMR comes into play, as it can contribute to aspects where X-ray crystallography and EM have limitations. Two strengths of solid-state NMR are of particular interest. First, conformational changes can be followed with highest sensitivity via the NMR chemical shifts (very small changes in protein conformation lead to considerable changes in resonance positions). Once the sequential assignment of a reference form is established, this allows to rapidly assess conformational variations as they occur, for instance, upon binding of an interaction partner to the reference structure. In fact, no full structure determination is needed to identify such conformational alterations, which makes the direct observation and quantification of structural transitions one of the strengths of NMR. The second major asset of NMR is its unrivaled ability to follow dynamic events over a wide range of experimental timescales that are too fast for cryo-EM and X-ray approaches, i.e., from picoseconds to seconds. NMR works under near-physiological conditions, and does not need the commonly very low temperatures of X-ray or cryo-EM, where mixed states of a sample are frozen out. Hence defining structural dynamics by these techniques often requires the availability of stable mimics of two or more different states.

This review reports on the recent progress in the field, and intends to give an overview on the emerging possibilities and expected insights which can be gained from solid-state NMR.

## 2. Solid-State NMR

Solid-state NMR is a relative newcomer to the structural virology field, as sufficient spectral resolution and sensitivity to address complex proteins with atomic resolution have only recently been developed. Liquid-state (“in-solution”) and solid-state NMR in fact started out at the same time in the laboratories of Bloch and Purcell [20,21], and the spectra looked remarkably similar [20]. Historically, it was the development of sophisticated pulse techniques and multidimensional spectroscopy in the 1970s and 80s [22,23,24] which set the stage for protein structure determination by solution-state NMR [25]. Then, technical improvements in the homogeneity, stability and strengths of the magnetic fields applied dramatically increased resolution in the spectra of complex biomolecules for liquid-state NMR. Indeed, in solution the anisotropic interactions in the spin system are averaged by molecular tumbling and thus the “natural linewidth” is narrow (a few Hertz). This provides a high enough resolution in order to distinguish hundreds of resonances as they occur in medium-sized proteins. A natural limitation is set by the molecular tumbling which, today, restricts in solution NMR the maximum molecular mass for meaningful analyses of rigid proteins to roughly 50 kDa. Notably, this limit can be significantly pushed up by selective methyl-group protonation, which dilutes the dipolar coupling network; this allows, for example, investigation into the dynamics of proteins with a higher molecular weight, such as the 670 kDa proteasome [26].

In contrast, protein solid-state NMR was long plagued by much broader lines (around 50 kHz, and thus more than a thousand times broader than in solution), in particular for protons. This is due to the absence of molecular tumbling, which results in orientation-dependent anisotropic contributions to the chemical shifts, and the presence of a strong dipolar-coupling network (see Figure 1). One early conceptual break-through was the observation that mechanical rotation of the sample narrows the resonance lines. The strongest effect is found if the rotation axis is inclined from the magnetic field direction by the “magic” angle of *Θ* ≈ 54.7° [27] for which the second-order Legendre polynomial, P2(cos θ) equals zero. By this, magic-angle spinning (MAS) removes the anisotropic contribution to the chemical shift and the dipole couplings if the rotation frequency is roughly of the same value as the size of the interaction to be averaged. The spectra then become liquid-like. The strengths of the interactions depends on their type, as well as the nuclei involved. While ^13^C chemical shift anisotropies are, for instance, on the order of a few kHz, and already very low spinning frequencies are sufficient to average them, it is highly beneficial to spin even faster than 100 kHz to average the strong proton–proton dipolar interactions. These approaches are currently under development [28,29].

Practical realization took a long time until improvements in ^13^C-detected spectroscopy led to the first high-resolution protein spectra that finally could be assigned [30] and be used for the first protein structure determination by solid-state NMR [31]. Early, proton detection had been applied at moderate spinning frequencies (40–60 kHz) to proteins using extensive deuteration to reduce proton–proton dipolar interactions [32,33]. Only in 2014, MAS frequencies finally exceeded 100 kHz in protein spectroscopy [34]. Indeed, this increase in MAS frequencies substantially increased resolution in deuterated proteins, and allowed for high-resolution ^1^H-detected spectroscopy also on fully protonated solid proteins, through more efficient averaging of the strong ^1^H-^1^H dipolar couplings. Detecting on ^1^H spins (as is routinely done in solution-state NMR) is more sensitive than ^13^C detection, mainly through the higher gyromagnetic ratio of the proton when compared to the ^13^C. This brought an almost 100-fold increase in mass sensitivity [35], reducing sample requirements to below 1 mg of protein and thus opening new possibilities in applications to biomolecules.

Still, the solid-state linewidth is typically somewhat broader than in solution-state NMR, which results in a higher spectral crowding than in solution, and limits the protein size to be investigated. Thus, further improvements in MAS frequency (>100 kHz [29,30,31,32,33,34,35,36]) and magnetic field strength (>800 MHz or 18.8 Tesla; note that magnetic field strength is conventionally expressed in units of the proton resonance frequency) will be important for the study of larger proteins. Typically, going from 800 MHz to 1.2 GHz increases the signal-to-noise ratio by nearly a factor of 2. But already today, the combination of 100 kHz MAS, high magnetic fields (>800 MHz) (Figure 1) and improved sample preparation, e.g., by direct sedimentation [37,38], allows for high-resolution spectroscopy of large biomolecular assemblies like viral capsids or envelopes [35,39,40,41] if they show high local symmetry. If the protein-monomers forming the assemblies are structurally diverse, this results in the observation of multiple sets of NMR signals (see also below), and increases spectral crowding or line widths if the splittings are unresolved.

Solid-state NMR is thus a technique which can address structure and detailed conformation of proteins which are “not soluble”. This term must be understood in the broader sense, often simply meaning that they are involved in molecular interactions leading to larger superstructures, either with lipids for membrane proteins, or with each other, as in molecular assemblies such as molecular machines, protein fibrils, viral capsids or envelopes. “Solid” in this context means that the proteins do not show fast enough molecular tumbling (with correlation times of tens of nanoseconds) to be addressed by solution-state NMR. In a first approximation, therefore, the most attractive targets for solid-state NMR are large assemblies of many relatively small subunits, the latter being important to avoid excessive signal overlap in the spectra.

As mentioned, an additional important feature of solid-state NMR is its ability to characterize dynamic processes by observing relaxation times and exchange effects [42]. For an NMR spectroscopist, dynamics is usually associated with a timescale (the correlation time) and an amplitude (the order parameter), while structural virology dynamics is more associated with the volume of the conformational space for the protein. Dynamics on the pico- and nanosecond timescale (backbone and sidechain motion) can be characterized by relaxation measurements, while microsecond dynamics often leads to very efficient relaxation. In solid-state NMR spectra, dynamic entities (often loops but sometimes entire domains) may completely vanish from the spectrum as a consequence of such dynamics [43]. While limiting the possibilities for structure determination, the ability to assign a defined range of motional timescale to the corresponding residues provides valuable dynamic information.

As in solution-state NMR, the chemical shifts in MAS solid-state NMR give a fingerprint of a protein, which can be used to easily assess structural differences arising, for example, in the presence of interactants. This is of central importance in viral proteins, which often show multiple functions and thus also interactions, during which they change conformation to adapt to the specific role they need to fulfill at a specific point in the viral life cycle.

## 3. NMR Sample Preparation

### 3.1. The Need for Recombinant Sample Production

A major challenge, as for all structural techniques, is the isolation of appropriate virus-like or capsid-like particles (VLPs and CLPs, respectively) from cell culture, or to in-vitro reconstruct the viral protein and its interactants one seeks to investigate. As most solid-state NMR approaches rely on the availability of stable-isotope-labeled protein preparations (^2^H, ^13^C, ^15^N) [44], isolation of VLPs or CLPs is often impractical; for HBV capsids, for instance, this would require large-scale cultures of HBV-producing human hepatoma cells growing on isotope-labeled medium, and efficient separation methods. For bacterial viruses, this is much less challenging, and solid-state NMR has successfully been used to investigate Pf1 bacteriophages directly as sediments from ultracentrifugation [45,46]. However, the phage structure is composed mainly of a single protein, and the viral genome is difficult to observe under standard conditions. The DNA signals could be assigned using selective unlabeling of the protein [47,48] or dynamic nuclear polarization (DNP) [49,50], a technique providing signal intensity enhancement using transfer polarization from unpaired electron spins to nuclear spins at low temperatures (typically 100 K) although at the cost of limited spectral resolution (for a review on DNP see [51]).

Another limitation for the investigation of authentic viral particles directly by solid-state NMR is that, while biosafe probes have been designed for the investigation of amyloid proteins [52], they do not reach the safety provided by a biosafety level 3 laboratory often needed for the study of infectious material.

Another challenge is to obtain the amounts of material needed for NMR structural analysis. Until recently, solid-state NMR samples routinely comprised >20 mg of pure protein. With the advent of advanced proton-detection fast MAS techniques, proteins can now be analyzed in 0.7 mm and even 0.5 mm rotors which reach over 150 kHz MAS frequencies (Figure 1a), yielding well-resolved proton-nitrogen correlation spectra [29,34,35,36,53]. The concomitant reduction in sample sizes to the sub-milligram level enables the use of alternative production techniques to bacterial expression. Still, several hundred micrograms of pure protein are usually required, and the isolation of sufficient amounts of authentic viruses, moreover in a single, defined maturation state, would require truly large-scale mammalian cell cultures; complexity in vivo is often further increased by dynamic modifications such as phosphorylation/dephosphorylation events.

Hence NMR specimens are typically not entire virions from patients or cell culture, but rather recombinant purified proteins, or their assembly products such as capsids or envelopes, typically around 500 µg per sample, which are centrifuged directly out of a liquid phase as a sediment into the NMR rotor (Figure 1a). This technique yields good-quality spectra [37,38] and usually keeps the samples stable and hydrated for months if not years [54]. NMR experiments are usually performed near room temperature (−10 °C to 40 °C).

### 3.2. The Size of the Proteins and Their Assemblies Amenable to “High Resolution” Solid-State NMR

Solid-state NMR has a strength for assemblies that are large but consist of small, optimally identical, repeating units, as is true for many viral capsids and envelopes. Indeed, only a single set of signals is expected for a core protein, even if many equivalent copies (hundreds for helical symmetry, 60 for *T* = 3 and *T* = 4 icosahedral symmetry) make up the viral capsid under investigation. If there are N non-equivalent monomers in the unit cell, N peaks may be observed per nucleus, depending on the conformational differences. If they are sizeable, the different peaks are resolved; if they are smaller, they may lead to line broadening or have no effect. While these effects add to the information contents of the spectra [55] (see also below), they also increase spectral overlap, which can be detrimental for large proteins. Solid-state NMR does not require long-range order of the biological assembly, as long as the monomeric subunits feature a similar chemical environment at the atomic level. More complex assemblies containing larger and/or more different proteins however become increasingly difficult to study in native forms, since the complexity of NMR spectra increases with the sheer number of amino-acid residues present. Although solid-state NMR does not have the intrinsic size limitations of liquid-state NMR, where larger proteins give ever broader NMR lines, spectral overlap complicates analysis, such that objects of more than ~500 inequivalent amino acids are difficult to resolve.

### 3.3. Labeled Protein Production

An important point to be considered is that most NMR techniques rely on labeling of the proteins with NMR-active stable isotopes such as ^13^C and ^15^N, in some cases also ^2^H (^2^H is an NMR-active nucleus with spin 1, but is here not detected; rather it dilutes the dense ^1^H network to further reduce the distance-dependent ^1^H-^1^H dipolar couplings, in order to further narrow ^1^H lines). Stable isotope labeling can indeed be achieved in mammalian cell cultures [56], but at larger scales (see above) this rapidly becomes very costly, and is thus best achieved by employing high-yield systems for protein production.

As a result, solid-state NMR of viral proteins relies largely on recombinantly produced material, which under appropriate conditions can be a valid, high homogeneity model, as is often confirmed by EM and X-ray crystallography. *E. coli* expression is the major workhorse here, offering straightforward isotope labeling of proteins. Genuine *E. coli* toolboxes have been developed for several viruses, able to produce assemblies mimicking defined in vivo states, e.g., residue-specific phosphorylation (for example [57]) thus allowing the exploration of the conformational space of viral proteins. As demands in sample quantity are diminishing (see above), other production approaches have recently been developed, and a highly attractive compromise between expression in eukaryotic cells vs. *E. coli* is the use of eukaryotic cell-free systems, optimally one that combines high expression yields with an increased folding capacity and access to a range of post-translational modifications [58]. One such system is based on wheat germ cell-free protein synthesis (WG-CFPS) [40,41,59,60]. The high efficiency and selectivity of isotopic labeling constitutes a major asset of this system for NMR studies. Importantly, and in contrast to cell-based production, only the protein of interest is isotopically labeled [61], enabling sample preparation without extensive purification. This is of particular interest, for example, for the study of ribonucleoprotein complexes which form directly during the cell-free synthesis reaction. In addition, metabolic scrambling of labels between different amino acids is much lower than in *E. coli*, and can further be minimized by using transaminase inhibitors and glutamine synthase inhibitors [62,63]. WG-CFPS is particularly attractive for deuteration, often used to reduce proton linewidths [64,65], as in contrast to bacterial expression it does not require proton back-exchange, circumventing denaturation and refolding steps which could compromise the native folding of proteins. WG-CFPS indeed allows for the simple addition of deuterated amino acids to the translation reaction performed in H_2_O medium. Selective labeling, by adding only the desired labeled amino acid to the translation reaction, can efficiently be performed in WG-CFPS [59,62,63,66], and help to significantly reduce the complexity of NMR spectra, as shown, for instance, by the successful application to the NS4B protein of hepatitis C virus [60] (see below). Another complexity-reducing approach is segmental labeling, whereby only part of a protein sequence is isotopically labeled while the remaining parts remain invisible. This was successfully applied to HIV-1 in tubular assemblies [67] and in spherical CLPs formed by the capsid protein [68]. Other schemes can be used, with those where only one partner is labeled particularly relevant for interaction studies, where the non-isotopically-labeled partner(s) remain(s) invisible, and all information on changes is contained in the target protein spectra [69]. Alternatively, labeling protocols using selectively ^13^C-labeled glucose (1-^13^C or 2-^13^C) as a carbon source during protein expression [70] can be used to improve spectral resolution and determine intermolecular restraints, as demonstrated on molecular machines such as the type III secretion system needle [71]. This technique has not yet been employed for viral assemblies but could be useful to study contacts between capsids subunits.

High sample purity is required to reach maximal NMR signal strength, yet low-level contaminants (<5–10% of the total sample) will typically be invisible in the NMR spectra. 

## 4. NMR Protocols

### 4.1. Sequential Resonance Assignments

Once the proteins are labeled, 2D and 3D NMR spectra are recorded at an MAS frequency that depends on the rotor size used (typically 17 kHz for 3.2 mm rotors to 100 kHz for 0.7 mm ones). In multi-dimensional spectra, each dimension corresponds to the recording of an ^1^H, ^15^N or ^13^C magnetization’s free induction decay over time, which is then Fourier-transformed in order to yield a frequency-domain spectrum, ideally containing a peak for each ^1^H, ^15^N or ^13^C nucleus in the protein. Multi-dimensional spectra help to resolve overlap which would be very high in one-dimensional spectra. In these spectra, a residue is characterized by a “douple” and triple of resonance frequencies in the 2 or 3 dimensions, respectively, e.g., ^15^N,^13^CA,^13^CO intra-residual correlations. A combination of those nuclei with 3D spectra enables to create inter-residual connections by correlating, for example, the ^15^N of one residue with the ^13^CO/^13^CA of the preceding one. Such schemes thus connect the amide ^15^N to both directions of the protein backbone, and allow for sequential resonance assignments [53,72,73,74,75]. This eventually enables to associate an NMR signal or frequency to every atom in the protein. 

### 4.2. Chemical Shifts Reveal Structural Information

Once the NMR fingerprint is assigned (i.e., all signals and thus frequencies in the spectra are assigned to a nucleus), it is easy to follow conformational changes induced by the addition of a ligand or any other event, by simply following the chemical-shift differences and localizing them on a known 3D structure. Important examples are protein-nucleic acid complexes, where structural changes can in principle even be followed via the chemical shifts from both sides. Protein and RNA or DNA can either be labeled at the same time, e.g., when preparing a viral capsid in *E. coli* in media containing labeled amino acids and nucleic-acid building blocks, or be labeled separately if one can reassemble the capsid in vitro with labeled or unlabeled RNA, as it is the case for HBV capsids [76].

### 4.3. Structure Determination

Protein structure determination by solid-state NMR involves combining dihedral angle restraints derived from chemical shifts [77] with distance restraints derived from dipolar couplings. These restraints can then be used to computationally find a solution that fulfills as many of them as possible. The detailed outcome of this optimization depends on the initial condition [78,79,80]. The best solutions constitute the NMR “bundle”, which characterizes the possible range of conformations for the protein. Due to signal overlap, de novo structure determination from solid-state NMR data is currently limited to proteins of up to about 200 amino acids in length. Only a few viral protein structures have been determined de novo using solid-state NMR today.

### 4.4. Investigating Interactions

NMR can efficiently assess interactions with small molecules, which is extensively used in drug screening by solution NMR [81,82]. For solid-state NMR, high-throughput screening is not yet routinely available, but first compatible sample changers have been developed. Screening can thus be done, though with limited efficiency, providing access to the rich information of NMR chemical shifts and allowing a detailed analysis of interactions with small molecules by similar approaches as in solution. A particular strength of solid-state NMR in such applications is the availability of side-chain information directly from 2D ^13^C correlation spectra, which indicates which ^13^C nuclei are spatially nearby the interacting site.

## 5. Examples of Solid-State NMR Studies on Viral Proteins

Some of the first viral targets in MAS solid-state NMR were bacteriophages, serving as relatively simple model systems for NMR methods development. Major advantages were their often small coat proteins, which can be readily prepared in stable-isotope-labeled form, and the absence of an envelope, presenting a system containing just one principal structural element (reviewed by [83] and more recently by [84]). Studies of intact filamentous bacteriophages were described recently [50]. In parallel, solid-state NMR on membrane proteins progressed further, first on oriented samples [85] and then by MAS methods, and the study of small viral membrane peptides set the stage for the emerging analyses of much more complex samples such as membrane proteins from human viruses. Figure 2 shows a selection of human viruses and their proteins studied by solid-state NMR today (with red labels), aiming to gain an insight into their dynamics and structure, and their interactions with viral genomes as well as cellular membranes.

Several studies have addressed the human immunodeficiency virus 1 (HIV-1), with an emphasis on the capsid and gp41 fusion proteins (Figure 2a). Influenza virus, notably its M2 ion channel, has been the topic of several studies (Figure 2b), studies on measles virus nucleocapsids have also been initiated (Figure 2c), and more recently hepatitis B virus (HBV) proteins, both from the duck HBV model and the human virus (Figure 2d), have been addressed. In addition, non-structural membrane proteins from the hepatitis C virus (HCV), such as the p7 ion channel and NS4B, are challenging current subjects (Figure 2e). Below we summarize first the successes and then the remaining challenges, with a focus on proteins from pathogenic human viruses.

### 5.1. Viral Capsids

Viral capsids are superstructures formed by core protein(s) in complex with the viral genome. They usually represent assemblies with high local symmetries, often icosahedral, but they can also have irregular shapes. Their obvious function is to protect the viral genome, but they have often multiple roles in the viral life cycle. To fulfill these diverse functions, e.g., intracellular transport and trafficking, entry and egress from the host cell, they have structures that can change over time and on demand. Their overall shapes can best be addressed by cryo-EM, but the dynamics of their functional domains let them frequently escape detection by this method. Similarly, structural details can go undetected, since even the resolution achievable nowadays may be insufficient to clearly pinpoint subtle changes between states. 

#### 5.1.1. HIV Capsids

Solid-state NMR investigations of the HIV-1 capsid were initiated by Polenova and coworkers some years ago. The viral capsid comprises 1500 copies of the 26 kDa capsid protein (CA) and forms a characteristic cone-shaped cage around two copies of the viral genomic RNA (Figure 2a). The structural organization of the mature capsid was defined as a “fullerene cone” based on cryo-EM images [88] (Figure 3a,b). CA comprises an N-terminal domain (NTD) and a C-terminal domain (CTD), connected by a flexible linker, the hinge region. NMR spectra of HIV-1 CA, obtained and assigned using sediments of conical and spherical assemblies, formed the basis for structural and dynamic investigations by this and other groups (reviewed in [89] and [90]). NMR revealed that, despite the very different morphologies of the investigated particles, the individual CA proteins adopt very similar structures therein [91]. Probing dynamic behavior on the nano- to microsecond time scales through measurements of chemical-shift anisotropy (CSA) tensors revealed that the CA hinge region is flexible on the millisecond time scale (the chemical shift is not an isotropic quantity described by a number, but is mathematically described by a second-rank tensor, characterized by a 3 × 3 matrix. In the present context a tensorial interaction leads to a characteristic intensity pattern for each NMR resonance line describing the anisotropy and asymmetry of the interaction. The anisotropic part is, however, spun out under MAS to yield only the isotropic average, but can be reintroduced by carefully designed pulse sequences and reveals information on dynamics.)

This conformational flexibility was suggested to be essential for the CA protein to adopt multiple and slightly different conformations in the assembled capsid [92]. In a similar approach, Polenova and coworkers identified the interaction of CA with the host protein Cyclophilin A (CypA), which regulates HIV-1 viral infectivity through direct interactions with the viral capsid. ^1^H-^15^N and ^1^H-^13^C dipolar tensors and NMR peak intensities revealed that dynamics on the microsecond-to-nanosecond timescale in the CypA binding loop are important for HIV-1′s escape from CypA dependence [93] (again, the dipolar interaction is described by a second rank tensor, which is normally averaged but can be re-introduced to reveal dynamic averaging). More recently, NMR chemical shift and dipolar tensors were measured in HIV-1 CA tubular assemblies (Figure 3c,d). This revealed, in combination with molecular dynamics and theoretical calculations, a drastic decrease in dynamics for residues in the CypA-binding loop of the tubular assemblies upon interaction with CypA [94]. Moreover, the interaction of CA with the restriction factor TRIM5α has recently been described by such a combined approach [95]. In parallel, solid-state NMR studies in combination with EM by Tycko and coworkers have shown that CTD and NTD of CA tubular assembly samples are largely rigid and that they immobilize their general secondary and tertiary structure upon formation of tubes [96]. Later, the authors suggested a curvature generation mechanism [97,98], which would be determined by structural variations in three regions of CA: around the inter-domain linker region, in the segments involved in intermolecular NTD–CTD interactions, and in the C-terminal tail. The effects of an HIV-1 maturation inhibitor on CA was revealed to be rather subtle through the analysis of NMR chemical shifts and relaxation parameters [68].

#### 5.1.2. HBV Capsids

The HBV core protein has been accessible in recombinant form from early on, and its structure has been extensively investigated by EM and X-ray approaches [7,14,100,101,102]. Still, it needed advanced biochemistry to develop expression systems adequate for the production of the full-length form, Cp183 [76], which in contrast to the truncated Cp149 used in most structural studies, packs nucleic acids. The HBV capsid structure is dynamic, and is believed to change/adapt according to the different roles the protein has at different points in its viral life cycle. Its main function is to serve as the building block of the nucleocapsid shell surrounding the pgRNA and viral polymerase. Cp expressed in bacteria assembles into icosahedral particles closely resembling those seen in infected liver [101]. The predominant icosahedral *T* = 4 capsid contains 240 copies of Cp in 120 dimeric subunits, and comprises four groups of symmetrically non-equivalent subunits arranged as 60 AB and CD homodimers (Figure 3e). The 20 kDa core protein contains an N-terminal domain of 140 residues, required for the assembly into capsid, joined by a 9-residues linker to an arginine-rich C-terminal domain (CTD, residues 150 to 183) responsible for several essential functions, including pgRNA binding [103]. The dynamic structure of the CTD and its interactions with the viral RNA remains one of the blind spots in capsid investigations today. Furthermore, the hypothesized conformational changes of the capsid believed to signal readiness for envelopment [104], or alternatively the predicted single-strand blocking signal [105], have escaped detection as of today.

The HBV capsid has recently been subject to solid-state NMR studies on capsids isolated from *E. coli* cells where they auto-assemble (Figure 3f). The sequential ^1^H/^13^C/^15^N assignments have been established as a basis for further studies [35,106]. The NMR spectra yielded high-resolution NMR fingerprints (Figure 3g), with linewidths similar to those obtained from the HIV capsids, confirming that this type of symmetrical object is well in the scope of solid-state NMR. A comparison between the chemical shifts of the core protein in its dimeric and capsid states revealed the residues which structurally adapt upon shell assembly [106]. The slight non-symmetry (“quasiequivalence”) between the four subunits was reflected in the NMR spectra [55], in which multiple peaks are observed for ~20% of residues located at the dimer–dimer interfaces at the five- and six-fold symmetry axes (Figure 3g,h), as shown at the example of Ala137. Hence the high sensitivity of NMR chemical-shifts to structural variations allowed the exact pinpointing of the residues where subtle local conformational adaptations are needed for lattice formation, i.e., the molecular hinges [55].

As described above, the development of proton-detected fast MAS NMR spectroscopy beyond 100 kHz sample rotation has recently made it possible to study sub-milligram amounts of protein samples, and enabled proton assignments of HBV Cp, using less than 500 μg of capsids [35]. With the proton resonances assigned, this approach can now be used for further interaction studies, with cellular partners that are available in small quantities, as well as with antivirals like the capsid assembly modulators (CAMs, also called core protein allosteric modulators, CpAMs). CAMs are currently under pharmaceutical development for targeting viral nucleocapsid assembly [107,108]; they act by either promoting assembly of regularly appearing but genome-less capsids, or they induce the formation of aberrantly shaped irregular multimers, which are not functional in viral replication; notably, recent data suggest that this distinction is not always strict in that the concentration of a compound and the time it is in contact can affect the morphological outcome [109]. In this context, the use of cell-free synthesis systems for sample preparation is particularly interesting as assembly modulators can act directly on the capsid protein exiting from the ribosome, and hence in a preassembly state. The sub-milligram Cp183 capsids obtained through cell-free protein synthesis are fully sufficient for the smaller sample requirements of fast MAS NMR, and they can simply be sedimented into a 0.7 mm NMR rotor after purification on a sucrose gradient (Figure 4a) [40]. The capsids directly pack nucleic acids during cell-free synthesis, and only the full-length protein successfully assembles in the system. In the presence of a heteroaryldihydropyrimidine (HAP) CAM, the cell-free Cp183 capsids opened up completely (Figure 4b) [40], indicating suitability of this system for solid-state NMR analysis. These data show how the combination of WG-CFPS and solid-state NMR can open the way for a better understanding of the action of CAMs under conditions that more closely mimic the situation in a cell.

#### 5.1.3. Measles Virus Nucleocapsids

Solid-state NMR studies have also been well-initiated on the helical assemblies of Measles virus (MeV) nucleocapsids (Figure 5) formed by multiple copies of the nucleoprotein (N). N is composed of two globular domains, the N- and C-terminal domains, which together form a stable peanut-shaped nucleoprotein core of 400 residues (N_CORE_), responsible for self-assembly and RNA binding. In addition, N contains an intrinsically disordered tail domain of 125 residues (N_TAIL_) required for the interaction with the polymerase complex. For the study, N_CORE_ was also studied with N_TAIL_ removed to obtain cleaved nucleocapsids, which are more compact and regular than the intact nucleocapsids (Figure 5a). ^1^H-detected solid-state NMR experiments were recorded to compare intact and trypsin-cleaved MeV nucleocapsids [110], which showed no major structural reorganization as both spectra look similar (Figure 5b).

Sequential assignments for site-specific information were not obtained on this large protein. Still, comparing bulk T_1*ρ*_ relaxation times for both assemblies showed longer relaxation times for the cleaved capsid, indicative of higher local dynamics on a submicrosecond timescale in N_CORE_ of the intact capsids [110]. This dynamic information provides an interesting complement to the meanwhile solved near-atomic structure of the N_CORE_ nucleocapsids [112].

### 5.2. Viral Envelopes

Viral envelopes are often highly complex assemblies of lipid-embedded glycoproteins. Their structure remains difficult to access, as they often only exist in multimeric forms, or even keep their native structure exclusively in the assembled envelope. As bacterial expression is most often not possible, alternative expression methods have been developed. An interesting and medically relevant example are recombinant HBV subviral particles (SVPs) comprising just the small envelope protein S, which can efficiently be produced in yeast and serves as the active constituent of the HBV vaccine. During HBV infection, similar non-infectious SVPs, containing all three envelope proteins and host phospholipids, are secreted by infected cells in huge excess over virions [113,114]. Yeast expression of the SVPs is not compatible with NMR analyses, since isotope labeling is too costly as ^13^C methanol must also be fed during the long maturation steps. It has been shown early-on that the small HBV envelope protein (HBsS) can assemble in the wheat-germ cell-free system [115]. This system was recently used to assemble SVPs from the S envelope protein of the related duck hepatitis B virus (DHBV) [41] (Figure 6a,b), in a similar manner to the HBV capsids [40] (Figure 4a,b). The addition of mild detergent allowed for the production of milligram amounts of these integral membrane protein assemblies, and a simple sedimentation after sucrose gradient isolation enabled NMR sample preparation [41]. The SVPs showed a rather narrow distribution of diameters, close to those formed during HBV infection [116], and similar antigenic properties as SVPs isolated from DHBV-infected ducks. First two-dimensional ^1^H-^15^N solid-state NMR hNH spectra showed a number of isolated peaks with linewidths comparable with model membrane proteins (Figure 6a). Analysis of the chemical-shift distribution allowed the conclusion that their constituent subunits displayed mainly α-helical secondary structure [41]. With such suitable resolution and sensitivity, the duck envelope protein thus represents an attractive pilot model for the structural study of its more complex human homolog, for which no conditions for assembly have yet been reported.

### 5.3. Other Viral Membrane Proteins

Membrane proteins in general are challenging systems for structural studies because of their hydrophobic nature, and the phospholipid bilayer which forms their native environment. While solution-state NMR and X-ray crystallography was widely used on small viral membrane proteins in detergent, solid-state NMR has the advantage that the proteins can be investigated in lipids, which are closer mimics of native membranes. Solid-state NMR investigations were first initiated on small viral membrane proteins, also called “viral membrane-spanning mini-proteins” [117], often using oriented samples to obtain spectra [118]. In this context, HIV-1, influenza and parainfluenza virus fusion peptides of ~20 amino acids have been studied early on [119,120,121,122] to reveal the structure and orientation behavior of these fragments in lipid bilayers under conditions mimicking membrane fusion. Later, technical advances in sample preparation and NMR equipment enabled larger fusion peptides to be studied.

#### 5.3.1. HIV gp41 and Vpu

Several domains from the HIV gp41 protein were thus investigated [123,124,125,126]. From these data, a topology model of gp41 could be built, with the transmembrane domain well inserted into the lipid bilayer, and the other domains lying on the surface of one bilayer, such that gp41 bridges two lipid bilayers in a hemifusion-like state [126].

Beyond gp41, the 81 residue HIV-1 specific virus protein “u” (Vpu) accessory protein has been addressed by solid-state NMR [127,128,129]. Vpu represents an oligomeric integral membrane phosphoprotein with numerous biological functions. The structure of full-length Vpu was solved in proteoliposomes and displayed two helices in the cytoplasmic domain forming a U-shape (Figure 7a). NMR data also revealed that the length of the inter-helical loop and the orientation of the third helix vary with the lipid composition, demonstrating its flexibility and accessibility for potential interactions [127].

#### 5.3.2. Influenza M2 Channel

Other well-studied examples of viral membrane proteins using solid-state NMR are Matrix-2 proteins of Influenza A (AM2) and B (BM2) viruses, which are multifunctional proteins that act at several stages of the virus life cycle and have a proton channel function essential for viral replication [131]. Both proteins are activated by low pH to mediate virus uncoating. AM2 conducts protons with strong inward rectification, while BM2 conducts protons both inward and outward. The structure of their trans-membrane channel domain has been extensively characterized by NMR spectroscopy, X-ray crystallography, and other physical methods [131]. Solid-state NMR added important information on the structure of the AM2 proton channel, thanks to seven successive studies: the structure of the monomeric transmembrane region was solved in 2001 [132], followed by the structure of the full transmembrane domain in lipid bilayers, in free form [133] and in complex with amantadine [134,135,136], an antiviral able to block the M2 ion channel pore. Additional information was gained from the structures of the drug-resistant AM2 mutant S31N [137,138], showing that amantadine resistance of mutant S31N is caused by steric hindrance. More recently, the solid-state NMR structures of the BM2 channel in its open and closed conformation were solved in a phospholipid environment [130] (Figure 7b), revealing that the channel is more accessible to water at a low pH and that side chain dynamics are the essential driver of proton shuttling. Solid-state NMR structures were all solved in lipid bilayers, whereas most X-ray structures, e.g., of the transmembrane domain of AM2 in 2008 [139], were solved in detergents. Notably, for the transmembrane domain of BM2, it was shown that the binding to micelles may cause structural perturbations that do not reflect the structure in native lipid bilayers [140].

In addition to the resolution of protein structures at the atomic level, solid-state NMR can be used to measure membrane curvature and determine the binding site of a protein in the membrane using magnetically oriented bicelles and off-magic-angle spinning. This approach was successfully applied to the M2 protein [141], revealing that it mediates virus assembly and induces high membrane curvature. These experiments can be applicable to other proteins inducing membrane-curvature, including those involved in membrane trafficking, membrane fusion and cell division.

#### 5.3.3. HCV p7 and NS4B

All proteins from viruses in the *Flaviviridae* family, including HCV and Dengue, are membrane-bound. Not only the structural proteins, but also the non-structural proteins thus represent an important challenge to analyze their conformations. Two HCV proteins have been subject to solid-state NMR studies, p7 and NS4B.

P7 protein is a small, hydrophobic transmembrane protein that participates in viral assembly and release. This 63-amino acid protein is generally categorized as a viroporin because it can oligomerize as hexamers to form ion channels in host cell membranes. Its structure is made up of seven different sections, amongst which four are helical segments. Early solid-state NMR studies of p7 aligned in phospholipid bilayers, providing the tilt angles of two of these segments [142] and a topology model of the protein in this environment [143]. p7 contains two trans-membrane helices, while HIV-1 Vpu has only one. Comparison of the solid-state NMR structures of both viroporins revealed significant differences in the structures and dynamics of their internal loop and terminal regions [144]. In both cases, solid-state NMR was compared to solution-state NMR data obtained in micelles, and Opella and coworkers first identified the localization of the transmembrane domains of p7 and Vpu in lipid bilayers. Unfortunately, the full p7 structure in lipids could not yet be solved, as this would help to resolve contrasting data between different detergent-based solution NMR studies [145,146,147,148,149]; the membrane might actually play a central role in determining p7 structure and oligomerization.

Another HCV protein on which studies in lipids are underway is the nonstructural protein 4B (NS4B). NS4B is a 27 kDa alpha-helical integral membrane protein of 261 amino acid residues. While the NS4B 3D structure has not been solved yet, a topology model is available [150]; it proposes that membrane association of NS4B is mediated by both transmembrane domains in its central parts, and determinants for membrane association in the N- and C-terminal regions [151]. However, given its important role for HCV, the need for more detailed structural information on NS4B is evident. A solid-state NMR study combined with molecular dynamic calculations looked at the second amphipathic helix of NS4B (AH2), known to promote remodeling events that are essential in viral replication [152]. Using a synthetic peptide reconstituted into lipids, the authors showed that AH2 promotes the clustering of negatively charged lipids within the bilayer, facilitating the remodeling of the latter, and increasing the disassociation of AH2 oligomers. Yet, insight into the multiple functions of NS4B which might be governed by distinct membrane topologies and/or interactions with other viral and cellular proteins [151] would be greatly supported by detailed structural information on the full-length protein. Again, WG-CFPS represents an attractive alternative to the poor expression of NSB4 in bacteria. Indeed, allowing for the addition of various additives, including detergents, this system enabled the production of full-length, well-folded and soluble NS4B in milligram amounts, compatible with structural studies [153,154]. The detergent-solubilized NS4B protein was affinity-purified, and then reinserted into lipids to mimic the native membranous environment [59,67,155]. A technical issue in such a reconstitution experiment is finding the optimal lipid-to-protein ratio (LPR) for NMR samples, because the LPR needs to be high enough to ensure a well-folded protein, but as low as possible to maximize protein amounts in the NMR rotor [44]. Adapting recent fast reconstitution schemes [156] has allowed to screen different lipid compositions directly by NMR to optimize resolution [67]. This enabled the recording of 2D and 3D ^1^H-detected NMR spectra, both on fully and selectively labeled NS4B and to initiate sequential assignments [67] (Figure 8). Two protein segments could be assigned, one validated the structural model, while the other did not confirm the predicted helix. For further work, higher magnetic fields will be an essential ingredient to push the structural understanding of this complex protein.

## 6. Conclusions and Outlook

Solid-state NMR is advancing in the analysis of viral proteins, concentrating on targets and aspects thereof which are complementary to other high-resolution approaches. It can typically provide important information on dynamic aspects, and also efficiently screen multiple forms of proteins created under different conditions, be it polymorphism, multimerization state, or interaction with partners. A uniquely strong point is the detection of conformational differences with very high sensitivity yet without the need for full structure determination. The recent development of fast MAS and the concomitant use of high-sensitivity proton detection has reduced sample amounts by a spectacular factor of a hundred, paving the way towards studying difficult (membrane) proteins which are only accessible in small amounts. The sensitivity gain is also essential to characterize the amplitude and correlation times of motional processes by NMR relaxation measurements. Solid-state NMR is also able to fill in the missing pieces in the dynamic structures protein access in the context of the viral life cycle, through its high tolerance towards sample preparation conditions, without the need for crystals or symmetry. It can thus reveal unique information on dynamics, membrane-orientation or interactions with partners. As strikingly illustrated for electron microscopy by the resolution revolution made possible by new detector techniques, many advances in solid-state NMR are also linked to technological innovation. Key developments under way are even faster magic-angle spinning, and ever higher field magnets, with the concomitant enabling a gain in resolution and sensitivity. Together, they are bringing many viral proteins and their assemblies into the focus of NMR investigations. Particularly novel and promising is the emerging opportunity to now include viruses that do not form ordered symmetrical capsids, but rather less regular nucleoprotein structures, like for example the ribonucleoprotein complexes from the dengue virus or the hepatitis D virus. Another interesting new option is the combination of solid-state NMR with cryo-EM in a hybrid structural approach, as established, for example, for amyloid fibrils [157,158]. These developments open exciting new research avenues in the future.

## Figures and Tables

**Figure 1 viruses-12-01069-f001:**
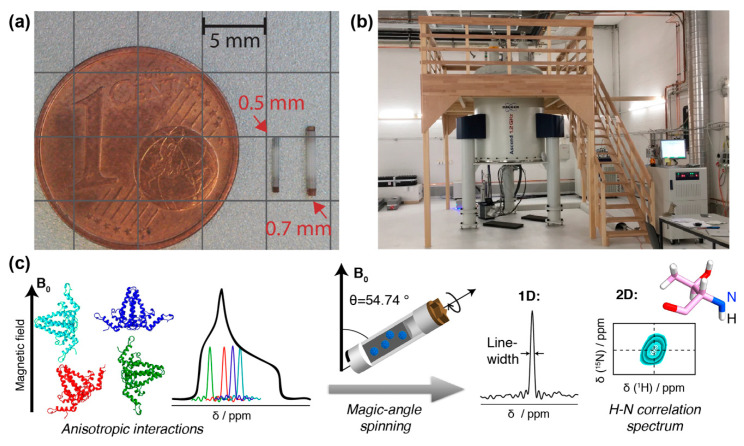
NMR rotors, spectrometer and resonance signals: (**a**) NMR rotors for fast spinning: 110 kHz for 0.7 mm rotor, 150 kHz for 0.5 mm (picture courtesy of Susanne Penzel). (**b**) 1200 MHz magnet for solid-state NMR at ETH Zurich. (**c**) Schematic representation of anisotropic interactions arising from different orientation of the individual molecules with respect to the magnetic field. The superposition of all possible chemical shifts gives rise to broad peaks with a characteristic shape, the powder pattern. The anisotropic interactions can be averaged out using MAS, which results in a single resonance line centered at the isotropic chemical shift of the spin. In addition, spinning sidebands may appear. This spin can be correlated to a neighboring spin, in the example, the amide ^1^H to the amide ^15^N. Two-dimensional spectroscopy then shows peaks which represent, for the present example, the amide proton frequency in one dimension, and the nitrogen frequency in the other. Such a signal will be observed for every NH pair in the protein. The resonance-line position is given by the isotropic part of NMR chemical shift and is usually specified in ppm (parts per million) of the resonance frequency. To obtain high-resolution spectra, the linewidth should be as narrow as possible, as this allows to distinguish (resolve) a maximum of resonances. A narrowing of linewidths can be achieved by (i) using a spectrometer operating at a higher magnetic field, (ii) improving sample homogeneity and symmetry, and (iii) reducing the dipolar coupling interactions by decreasing the density of protons in the system (through protein deuteration, for example) and/or increasing the MAS frequency by using smaller diameter rotors.

**Figure 2 viruses-12-01069-f002:**
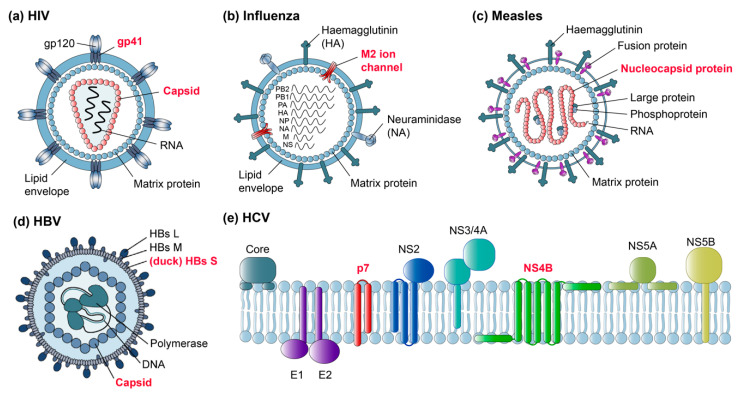
Overview of human viruses where solid-state NMR contributed to the understanding of global organization, orientation in the membrane and protein structure and dynamics. Viral proteins investigated by solid-state NMR are labeled in red. (**a**) HIV-1 virion with envelope-embedded gp41 protein containing the fusion peptide and internal capsid. (**b**) Influenza virion with M2 ion channel. (**c**) Measles virion with nucleocapsid protein. (**d**) HBV virion with small surface protein S and internal capsid. (**e**) Membrane topology of HCV with p7 and NS4B membrane proteins. Panel (**d**) is reprinted from [86], Copyright (2014), with permission from Elsevier. Panel (**e**) is adapted from [87].

**Figure 3 viruses-12-01069-f003:**
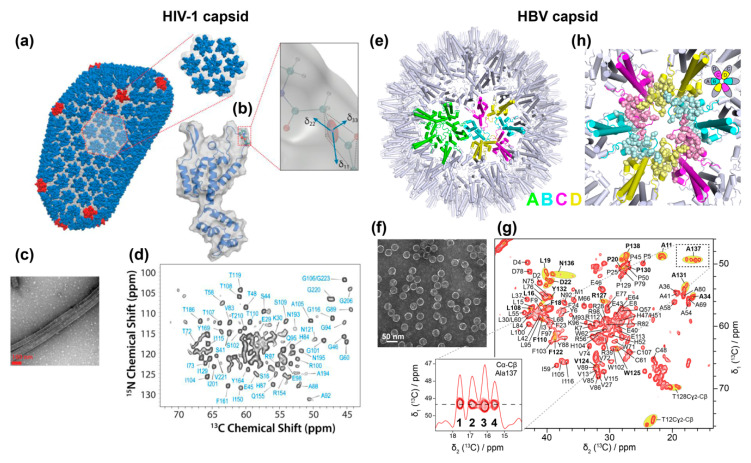
Solid-state NMR studies on the HIV and HBV capsids. (**a**) Illustration of the mature HIV-1 capsid (PDB: 3J3Y [99]); and the CA protein in (**b**), with a zoom on the backbone ^15^N^H^ CSA tensor and its orientation in the protein molecular frame. (**c**) Transmission EM image of CA and (**d**) the corresponding 2D NCA spectrum recorded at 14 kHz MAS. (**e**) HBV capsid structure (PDB: 1QGT [7]) showing the pentamers formed by A subunits and hexamers formed by B, C and D subunits. (**f**) Negative-staining EM picture of HBV capsids and (**g**) corresponding 2D Dipolar Assisted Rotational Resonance (DARR) spectrum recorded at 17.5 kHz MAS. Residues with NMR peak splitting due to the asymmetric subunits are highlighted in yellow and are shown as spheres in (**h**) on the capsid structure. Pictures from panels (**a**–**d**) were reprinted with permission from reference [94], Copyright (2016) American Chemical Society. Pictures from panels (**e**–**h**) were taken with permissions from reference [55].

**Figure 4 viruses-12-01069-f004:**
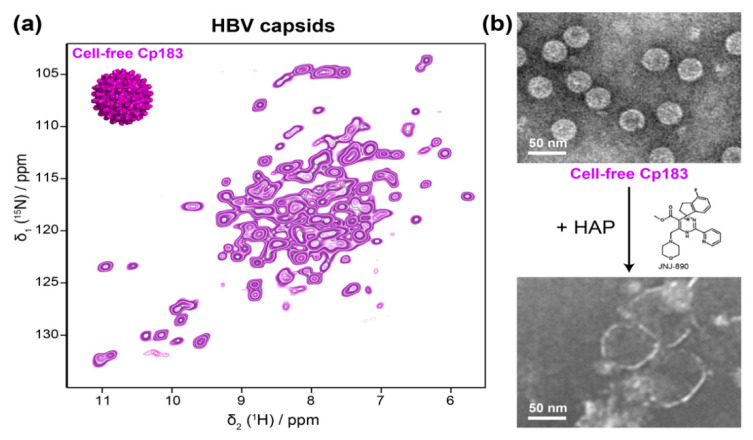
(**a**) 2D hNH spectrum of HBV capsids made of full-length core protein (Cp183) expressed and auto-assembled using WG-CFPS, recorded at 100 kHz. (**b**) Negative-staining EM pictures of the capsids prepared by WG-CFPS in the absence (top) and in the presence (bottom) of a JNJ-890 capsid assembly modulator, leading to opened objects. Figure adapted with permissions from reference [40].

**Figure 5 viruses-12-01069-f005:**
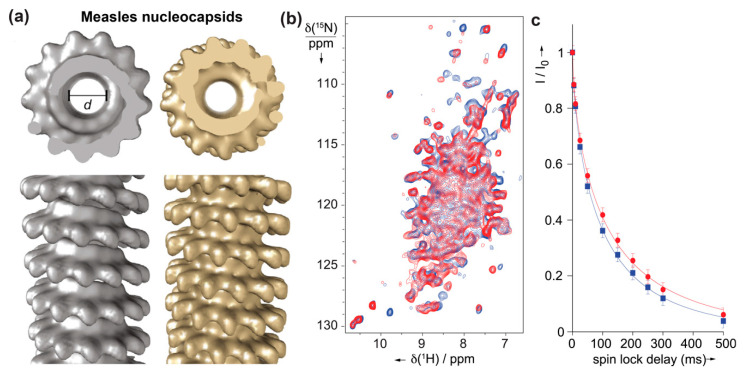
(**a**) EM pictures of intact (grey) and cleaved (brown) measle nucleocapsids [111]. (**b**) 2D hNH spectra recorded at 60 kHz MAS of intact (blue) and cleaved (red) measle nucleocapsids and corresponding bulk ^15^N R1rho decays in (**c**), revealing longer life times and thus different dynamics for the cleaved form. Figure reprinted from [110], Copyright (2014), with permission from Elsevier.

**Figure 6 viruses-12-01069-f006:**
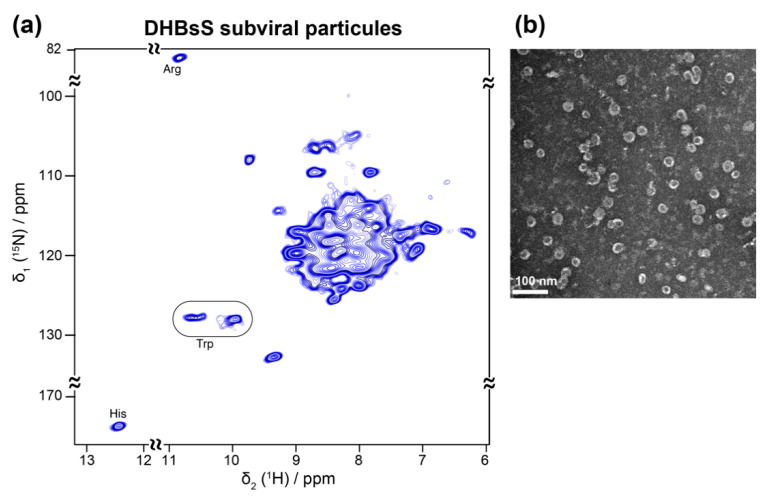
(**a**) 2D hNH spectrum of DHBs small envelope protein produced using WG-CFPS, recorded at 110 kHz MAS. (**b**) Negative-staining EM pictures of the corresponding protein, which self-assembles into spherical particles of ~29 nm. Figures taken with permission from reference [41].

**Figure 7 viruses-12-01069-f007:**
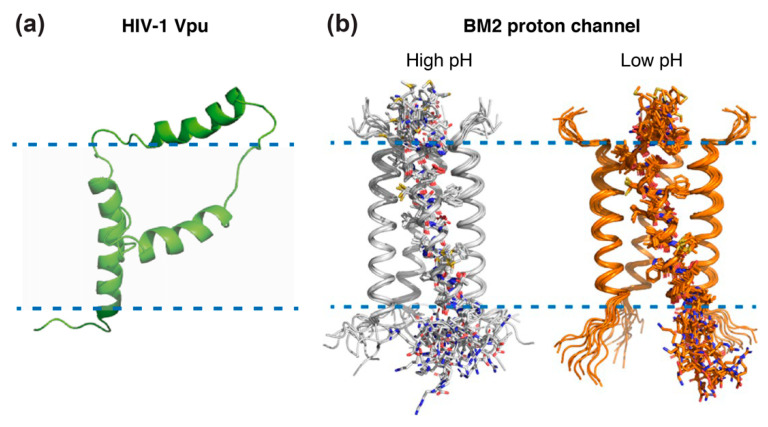
Solid-state NMR structures of (**a**) Vpu from HIV-1 (PDB: 2N28 [127]), and (**b**) closed and open influenza BM2 channels in lipid membranes (PDB: 6pvr (pH 7.5) and 6pvt (pH 4.5) [130]). Panel (**a**) reprinted from reference [127] Copyright (2015), with permission from Elsevier. Panel (**b**) reprinted from reference [130] Copyright (2020), with permission from Springer Nature.

**Figure 8 viruses-12-01069-f008:**
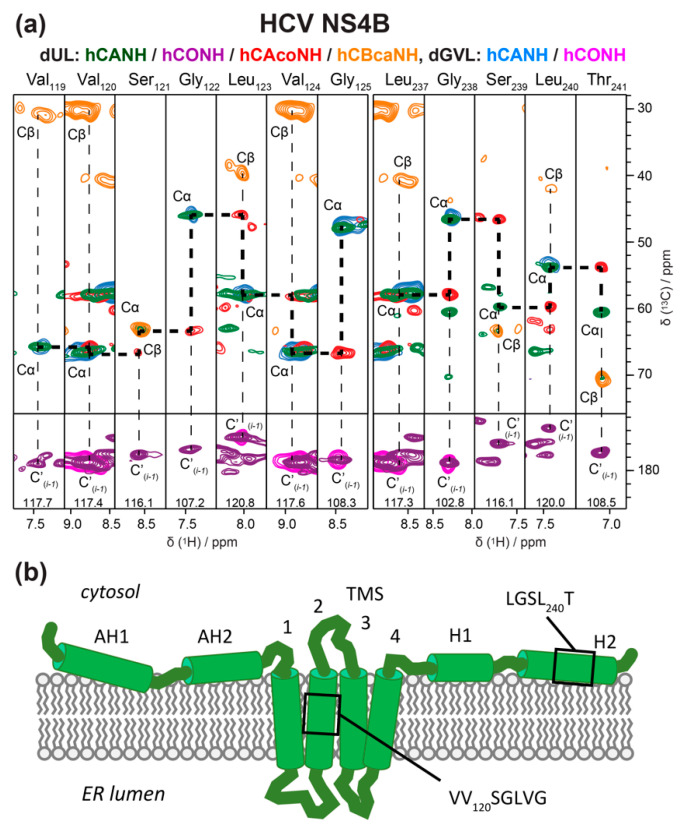
(**a**) 3D correlation spectra of deuterated NS4B from HCV with sequential connectivities. Selected strip plots representing the assignment of amino acid stretches of residues 119 to 125 and 237 to 241 using a set of 6 spectra recorded at 110 kHz MAS on two NS4B samples with different labeling schemes [60]. The orange spectrum was acquired at 60 kHz MAS. (**b**) A putative topology model of the NS4B protein adapted from [150], in which NS4B is proposed to contain four presumably amphipathic α-helices and four predicted transmembrane segments in the middle. The black boxes indicate the location of the two assigned regions shown in panel a. Figure reproduced with permission from reference [60].

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
