# Peer review of "Solid-State NMR for Studying the Structure and Dynamics of Viral Assemblies"

_viruses, 2020, doi:10.3390/v12101069_

Round 1

Reviewer 1 Report

This is a timely review dealing with the utilization solid-state NMR (SSNMR) to study viruses, with emphasis on HIV, HBV, HCV, Measles and Influenza. The choice has been focused on viruses associated with various human diseases. The studies represent various aspects of the power of SSNMR to provide detailed information on their structure and dynamics. The introduction is well designed to present the method to the non-expert reader with emphasis on recent hardware breakthroughs in the field of SSNMR.  The authors describe applications to study viral capsid (within the framework of VLPs – virus-like-particles) prepare in-vitro with some interesting and very useful expression system (Wheat-Germ Cell-Free expression), they describe studies of nucleocapsids (Measles), viral envelopes and membrane-associated viral peptides. There are really only a few minor comments that are described below:

First paragraph of Section 2 – the authors discuss the limitations of solution NMR and indicate the size limit of 50 kDa. I think at least one sentence should be mentioned on the study of molecular machines by solution NMR which has been pioneered by Lewis Kay and co-workers where methyl labeling allows to study proteins of very high molecular weights in solution.

Third paragraph of Section 2 – Here the authors discuss how 1H MAS SSNMR became possible with the invention of fast spinning. They should mention in brief also 1H NMR enabled by deuteration at low spinning speeds as shown by Rief, Zilm, Linser and others. For example (but not only),  JACS 136, 11002, 2014.

First paragraph of section 3 – the authors discuss briefly SSNMR applications to study bacteriophage viruses. In line 183 they state how Pf1 could be investigated directly from sedimentation with what seems to be mentioning the first application of NMR to study whole viruses. Precipitated Pf1 bacteriophage has been studied by McDermott already in 2007 (JACS 129, 2338). In line 185 they mention that the genome could not be studied under standard conditions. I am not sure what “standard” means here but the 31P of the DNA of Pf1 has been observed Opella, 13C/15N data was obtained by McDermott (JACS 133, 20208, 2011), and interactions with the capsid have been observed in the group of Goldbourt for fd (JACS 136, 2292, 2014).

First paragraph of section 3, line 194 – the authors mention up to 60 identical copies for icosahedral symmetry. Isn’t it 180 arranged in dimers or trimers?

Line 223 in page 5 – “Contrary to cryo-EM, solid-state NMR …”. I belive this should by X-ray and not cryo-EM. Cryo does not require long-range order.

End of this section, page 6 – The authors mention a preparation protocol of segmental labeling. Perhaps worth mentioning also the preparation of mixed labels, for example as done by Lange and co-workers for secretion needle tubes (1-glucose and 2-glucose). Indeed this was not applied to viruses yet but surely will as it presents a way to look at contacts between capsid subunits (e.g. Acc. Chem. Res. 2013, 46, 9, 2070–2079 or the reference within).

Section 4, references to sequential resonance assignments for N/C-based 3D (43, 61, 62). This goes back to Rienstra (JACS 124, 10979, 2000) and Oschkinat (ChemBioChem 2, 272, 2001).

Page 7, “de-novo structure determination … limited to … 100-200 amino acids”. Why not “up to about 200 amino-acids? This looks like excluding smaller proteins. Also, examples exist for larger proteins (e.g. Ladizhansky TM7). This is of course a unique example.

Author Response

 First paragraph of Section 2 – the authors discuss the limitations of solution NMR and indicate the size limit of 50 kDa. I think at least one sentence should be mentioned on the study of molecular machines by solution NMR which has been pioneered by Lewis Kay and co-workers where methyl labeling allows to study proteins of very high molecular weights in solution.

>> A sentence was added with references.

Third paragraph of Section 2 – Here the authors discuss how 1H MAS SSNMR became possible with the invention of fast spinning. They should mention in brief also 1H NMR enabled by deuteration at low spinning speeds as shown by Rief, Zilm, Linser and others. For example (but not only),  JACS 136, 11002, 2014.

>> A sentence was added with references.

First paragraph of section 3 – the authors discuss briefly SSNMR applications to study bacteriophage viruses. In line 183 they state how Pf1 could be investigated directly from sedimentation with what seems to be mentioning the first application of NMR to study whole viruses. Precipitated Pf1 bacteriophage has been studied by McDermott already in 2007 (JACS 129, 2338). In line 185 they mention that the genome could not be studied under standard conditions. I am not sure what “standard” means here but the 31P of the DNA of Pf1 has been observed Opella, 13C/15N data was obtained by McDermott (JACS 133, 20208, 2011), and interactions with the capsid have been observed in the group of Goldbourt for fd (JACS 136, 2292, 2014).

>> The paragraph was modified to include the suggested references and now mention the studies on the Pf1 DNA.

First paragraph of section 3, line 194 – the authors mention up to 60 identical copies for icosahedral symmetry. Isn’t it 180 arranged in dimers or trimers?

>> Not to our knowledge – the icosahedron consists of 60 asymmetric units, and an icosahedral virus is made of 60N protein subunits. For the HBV capsid, this yields 180 monomers for the T=3 and 240 monomers for the T=4.

Line 223 in page 5 – “Contrary to cryo-EM, solid-state NMR …”. I belive this should by X-ray and not cryo-EM. Cryo does not require long-range order.

>> We deleted.

End of this section, page 6 – The authors mention a preparation protocol of segmental labeling. Perhaps worth mentioning also the preparation of mixed labels, for example as done by Lange and co-workers for secretion needle tubes (1-glucose and 2-glucose). Indeed this was not applied to viruses yet but surely will as it presents a way to look at contacts between capsid subunits (e.g. Acc. Chem. Res. 2013, 46, 9, 2070–2079 or the reference within).

>> A sentence was added at the end of section 3 to include this suggestion.

Section 4, references to sequential resonance assignments for N/C-based 3D (43, 61, 62). This goes back to Rienstra (JACS 124, 10979, 2000) and Oschkinat (ChemBioChem 2, 272, 2001).

>> We added the two references.

Page 7, “de-novo structure determination … limited to … 100-200 amino acids”. Why not “up to about 200 amino-acids? This looks like excluding smaller proteins. Also, examples exist for larger proteins (e.g. Ladizhansky TM7). This is of course a unique example.

>> The sentence was modified with ‘up to 200’ instead of ‘about 100-200’. The structure determined using solid-state NMR by Ladizhanski was actually previously known from x-ray (L. Vogeley, O. A. Sineshchekov, V. D. Trivedi, J. Sasaki, J. L. Spudich, H. Luecke, Science 2004, 306, 1390).

Reviewer 2 Report

Your review describes the current state-of-the-art of using solid state NMR approaches to investigate the structure and dynamics of viral assemblies. You very thoroughly and extensively describe examples of the successful use of solid state NMR to investigate viral proteins, but in my view the introduction parts are too technical and NMR-terminology specific to be useful for someone outside of the field, especially if they do not have a structural biology background. I therefore suggest the following improvements:

- From line 86- onwards you start introducing solid state NMR and refer to ‘resonance positions’ without any prior introduction. This referring to technical terms (‘anisotropic interactions’, ‘line width’, ‘resolution’, …) continues throughout the text without introducing them, and is therefore not meaningful without some understanding of NMR. I suggest that, instead of showing Figure 1, which is not really useful at all in giving any kind of insight into how NMR works, you show an overview figure that illustrates the main concepts that you use in your review; how does a resonance relate to an atom, what is the line width then within a spectrum, what is anisotropy/isotropic, could all be visualised and give the non-initiated reader at least a concept of what this all means. 

- There are rather random facts distributed throughout the first sections, for example is it really necessary to go into detail about the use of RMSD in lines 45-48? This is a technical point not suitable for a general introduction. Please check your text for other such occurrences, and please keep in mind that your audience is not necessarily NMR-initiated.

- You might want to give an indication of what the introduction paragraphs cover, to give the subsections more structure. E.g. in section 3, you could group together the information by topic, for example bio-hazards, sample purity, … this will make it easier for the reader to access the information.

- You might want to include a section on the importance of computational methods for data interpretation (section 4), and refer to hybrid structure determination methods where solid state NMR data is used alongside X-ray, EM, solution NMR data (section 6).

Other comments:

- There are some odd phrasings through the text, for example  ’devoid of independent life’ (in abstract) should be ‘incapable of independent life’. In particular, please change all occurrences of ‘stronghold’ (`1. a well-fortified place; fortress. 2. a place that serves as the center of a group, as of militants or of persons holding a controversial viewpoint:’) to ‘strengths’. A re-read by a native speaker colleague would resolve these issues quickly.

Author Response

- From line 86- onwards you start introducing solid state NMR and refer to ‘resonance positions’ without any prior introduction. This referring to technical terms (‘anisotropic interactions’, ‘line width’, ‘resolution’, …) continues throughout the text without introducing them, and is therefore not meaningful without some understanding of NMR. I suggest that, instead of showing Figure 1, which is not really useful at all in giving any kind of insight into how NMR works, you show an overview figure that illustrates the main concepts that you use in your review; how does a resonance relate to an atom, what is the line width then within a spectrum, what is anisotropy/isotropic, could all be visualised and give the non-initiated reader at least a concept of what this all means. 

>> We extended the Figure.

- There are rather random facts distributed throughout the first sections, for example is it really necessary to go into detail about the use of RMSD in lines 45-48? This is a technical point not suitable for a general introduction. Please check your text for other such occurrences, and please keep in mind that your audience is not necessarily NMR-initiated.

>> We removed.

- You might want to give an indication of what the introduction paragraphs cover, to give the subsections more structure. E.g. in section 3, you could group together the information by topic, for example bio-hazards, sample purity, … this will make it easier for the reader to access the information.

>> Subsections were added in section 3 and 4.

- You might want to include a section on the importance of computational methods for data interpretation (section 4), and refer to hybrid structure determination methods where solid state NMR data is used alongside X-ray, EM, solution NMR data (section 6).

>> We added in computational methods for structure determination, and an outlook on hybrid approaches where suggested.

Other comments:

- There are some odd phrasings through the text, for example  ’devoid of independent life’ (in abstract) should be ‘incapable of independent life’. In particular, please change all occurrences of ‘stronghold’ (`1. a well-fortified place; fortress. 2. a place that serves as the center of a group, as of militants or of persons holding a controversial viewpoint:’) to ‘strengths’. A re-read by a native speaker colleague would resolve these issues quickly.

>> Thanks - we replaced.

Reviewer 3 Report

The review provides a complete and remarkable overview on the use of solid-state NMR for studying molecular protein assemblies, describing the state-of-the-art approaches currently applied for the characterization of viral proteins. The review is well written and can be published in the present form.

Author Response

NA